# PhaForce: Phase-Scheduled Visual–Force Policy Learning with Slow Planning and Fast Correction for Contact-Rich Manipulation

*Abstract*—Contact-rich manipulation requires not only vision-dominant task semantics but also timely closed-loop reactions to force/torque (F/T) transients. Yet, generative visuomotor policies are typically constrained to low-frequency updates due to inference latency and action chunking, underutilizing F/T for control-rate feedback. Furthermore, existing force-aware methods often inject force continuously and indiscriminately, lacking an explicit mechanism to schedule *when*, *how much*, and *where* to use force across different task phases. We propose PhaForce, a phase-scheduled visual–force policy that coordinates low-rate chunk-level planning and high-rate residual correction via a unified contact/phase schedule. PhaForce comprises (i) a contact-aware phase predictor (CAP) that estimates contact probability and phase belief, (ii) a Slow diffusion planner that performs dual-gated visual–force fusion with orthogonal residual injection to preserve vision semantics while conditioning on force, and (iii) a Fast corrector that applies control-rate phase-routed residuals in interpretable corrective subspaces for within-chunk micro-adjustments. Across five real-robot contact-rich tasks, PhaForce achieves an average success rate of 86% (+40 pp over baselines) while also improving contact quality and robustness under OOD geometric shifts.

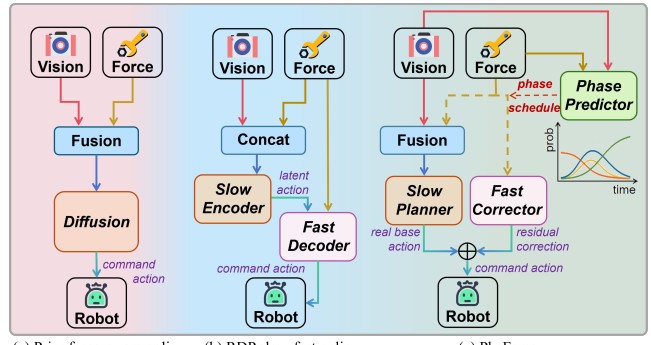

Fig. 1. Comparison of force-aware policy architectures. Prior works fuse vision and force into a single generative policy, while **PhaForce** introduces an explicit contact/phase schedule to coordinate chunk-level planning and within-chunk correction.

## I. INTRODUCTION

Diffusion-based visuomotor policies [1] and recent VLA models [2]–[4] have shown strong performance on vision-dominant manipulation tasks such as pick-and-place, rearrangement, and folding [5], [6]. However, many real-world skills are inherently *contact-rich*: success depends not only on geometric alignment but also on interaction dynamics such as friction, jamming, and transient impacts [7]–[12]. In insertion, being fully seated versus jammed on the rim can be visually indistinguishable at millimeter scale, while wrench transients reveal misalignment and recovery cues [13]. In wiping, images rarely reveal whether the tool is detached or over-pressed [11].

This motivates incorporating F/T sensing as physical feedback for contact-rich manipulation. Most force-aware policies encode a short F/T history and fuse it with vision, then use the fused representation in a chunked generative policy [14]–[17]. However, two structural gaps remain. **Gap 1: timescale mismatch of force feedback.** F/T is most useful as rapid closed-loop feedback, while generative policies are typically constrained to low-frequency updates by inference latency and action chunking. When force is primarily consumed at the chunk update rate, short-horizon transients such as stick–slip, micro-impacts, and early jamming can be under-reacted. **Gap 2: lack of explicit phase scheduling.** Reactive designs improve within-chunk reactivity [18], but contact-rich manipulation is inherently multi-phase. Different phases, such as planar search, insertion, unlocking,

or wiping, demand different corrective subspaces. Without an explicit phase schedule, high-rate reactivity can introduce spurious corrections in irrelevant dimensions, degrading alignment and causing failures.

We propose **PhaForce**, a *phase-scheduled* slow–fast policy that uses contact probability and task-defined phase belief to coordinate both low-rate planning and high-rate correction. **PhaForce** contains three components: (1) a *contact-aware phase predictor* (CAP) that outputs contact probability and a soft phase distribution; (2) a **Slow** diffusion planner that performs dual-gated visual–force fusion with orthogonal residual injection (ORI), so that force informs planning without overwriting vision-dominant semantics; and (3) a **Fast** residual corrector that applies control-rate corrections in phase-routed corrective subspaces. Our contributions are three-fold: (1) a phase-scheduled slow–fast policy that unifies force-aware chunk-level planning and control-rate residual correction; (2) an explicit scheduling signal—contact probability plus phase belief—that decides *when / how much / where* force should be used; and (3) real-robot validation on multiple contact-rich tasks, showing improved success rate, contact quality, and OOD robustness.

## II. METHOD

We formulate contact-rich manipulation as a slow–fast policy pair $(\pi_{\text{slow}}, \pi_{\text{fast}})$ under action chunking. The planner observation $o_t^p = (\mathcal{I}_t, w_t^{\text{hist}}, s_t)$ contains multi-view RGB, wrench history, and proprioception, while the corrector observation $o_t^c = (w_t^{\text{hist}}, s_t)$ excludes images and uses only low-dimensional signals. The **Slow** planner runs at inference rate $f_s$, whereas the **Fast** corrector runs at control rate $f_c$.

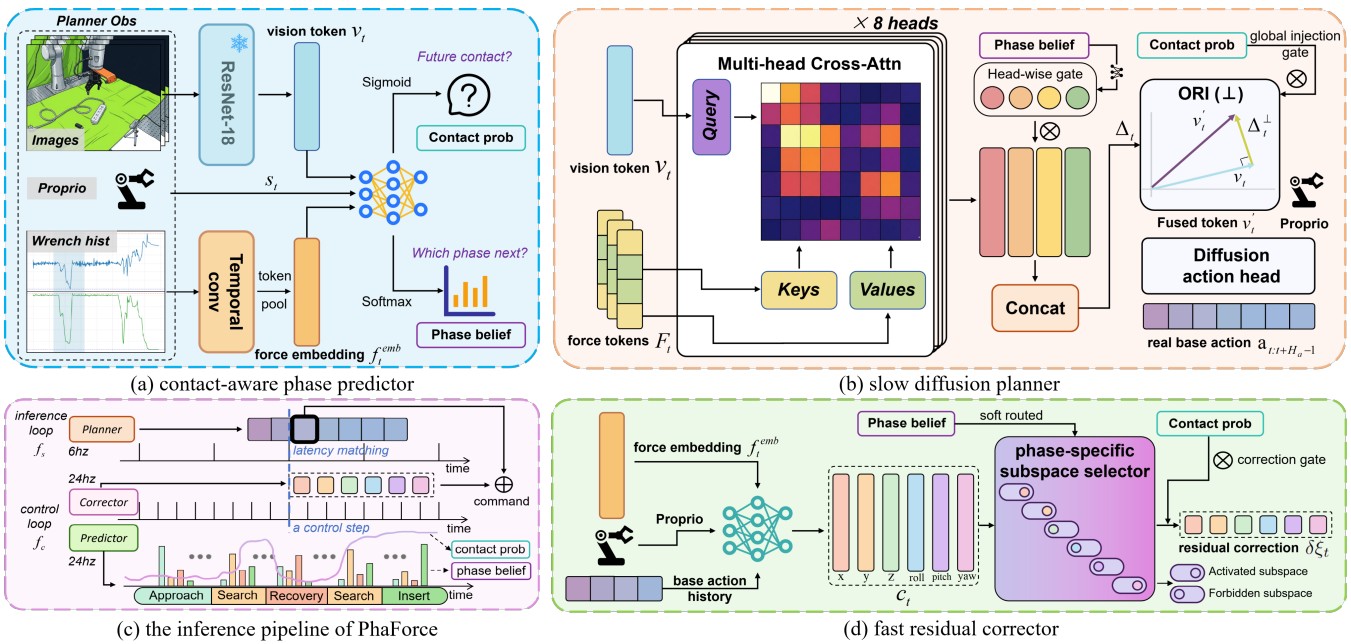

(a) contact-aware phase predictor

(b) slow diffusion planner

(c) the inference pipeline of PhaForce

(d) fast residual corrector

Fig. 2. **PhaForce architecture.** The **Slow** diffusion planner runs at $f_s$=6 Hz to generate action chunks, while **CAP** and the **Fast** corrector run at the control rate $f_c$=24 Hz for contact/phase prediction and within-chunk correction. In **Slow**, dual-gated vision–force fusion with ORI preserves vision-dominant semantics. In **Fast**, phase-belief soft routing activates corrective subspaces and produces a residual correction composed with the Slow base action.

If $T_t^{\text{slow}} \in SE(3)$ is the nominal pose from the planner and $\Delta T_t^{\text{fast}} \in SE(3)$ is the residual from the corrector, the executed pose is

$$T_t = T_t^{\text{slow}} \circ \Delta T_t^{\text{fast}}. \tag{1}$$

### A. Contact-aware phase predictor

Beyond binary contact state, contact-rich tasks are naturally multi-phase. For each task we define $K$ task-specific phases and train a lightweight predictor **CAP** to output a contact probability $p_t^c \in [0, 1]$ and a phase belief $\mathbf{p}_t \in \Delta^{K-1}$. CAP is trained with future-contact and future-phase supervision:

$$\mathcal{L}_{\text{CAP}} = \mathcal{L}_{\text{BCE}}(y_t^c, \ell_t^c) + \lambda_\phi \mathcal{L}_{\text{CE}}(y_t^{\text{phase}}, \ell_t^\phi). \tag{2}$$

Crucially, CAP is anticipatory rather than instantaneous, so the schedule can trigger search, insertion, or recovery before failure fully materializes.

### B. Slow diffusion planner

The **Slow** planner receives $(o_t^p, p_t^c, \mathbf{p}_t)$ and predicts an executable action chunk. We encode multi-view RGB into a visual token $v_t$ and wrench history into force tokens $F_t$. Vision and force are fused by multi-head cross-attention, using the visual token as query. To make force usage phase-dependent, we introduce a head-wise gate

$$g_t^{\text{head}} = \sigma(\text{MLP}(\mathbf{p}_t)), \tag{3}$$

which reweights attention heads according to phase belief, while a global contact gate $g_t^c = p_t^c$ controls the overall injection strength. To preserve vision-dominant semantics, we do not directly overwrite $v_t$ with the fused feature $\Delta_t$.

Instead, we keep only its orthogonal component and inject it residually:

$$\Delta_t^\perp = \Delta_t - \frac{\langle \Delta_t, v_t \rangle}{\langle v_t, v_t \rangle + \epsilon} v_t, \quad v_t' = v_t + \alpha g_t^c \Delta_t^\perp. \tag{4}$$

This ORI mechanism prevents semantic drift while allowing force to influence long-horizon planning when contact becomes relevant.

### C. Fast residual corrector

The **Fast** corrector receives $(o_t^c, \mathbf{h}_t^{\text{slow}}, p_t^c, \mathbf{p}_t)$ and outputs a 6D channel-wise residual $c_t$. For each phase $k$, we define a diagonal binary mask $B_k$ that selects admissible correction channels. For example, plug-in search mainly activates $(x, y, \text{yaw})$, while wiping mainly activates $z$. The executed residual twist is obtained by phase-belief soft routing:

$$\delta \xi_t = p_t^c \left( \sum_{k=1}^{K} \mathbf{p}_t^{(k)} B_k \right) c_t. \tag{5}$$

This yields interpretable, phase-routed corrections and suppresses spurious actions in free-space. We supervise **Fast** with phase-wise physical priors derived from wrench signals, such as relieving tangential friction in search or tracking desired normal force in wiping. Overall, the same contact/phase schedule coordinates both long-horizon planning and high-rate correction.

## III. EXPERIMENTS

### A. Setup

Experiments are conducted on a Flexiv Rizon 4s arm with a 6-axis force/torque sensor, one wrist-mounted and

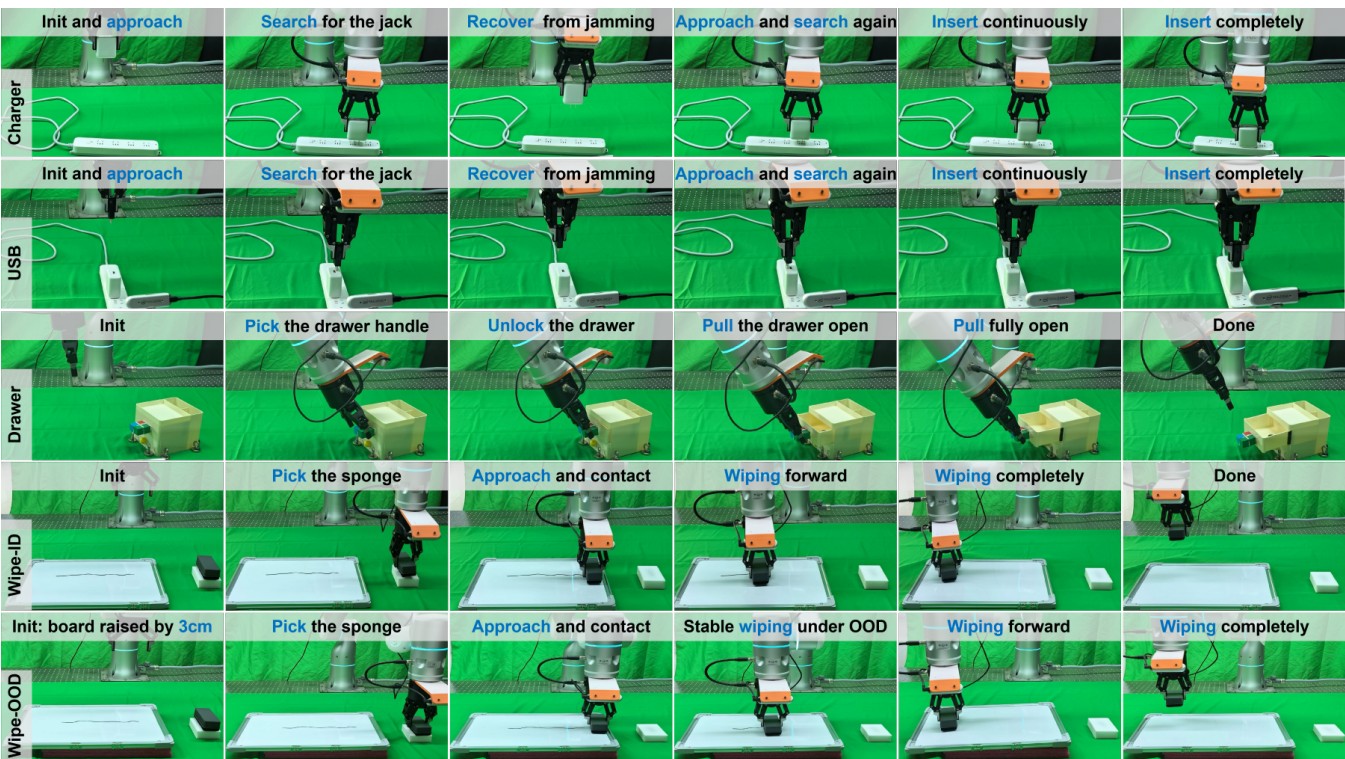

Fig. 3. Qualitative results on five contact-rich tasks. Each row illustrates a representative phase transition sequence. **PhaForce** handles search, recovery, compliant pulling, and stable wiping, and remains effective under OOD geometric shift.

TABLE I
SUCCESS RATE (SR, %) ACROSS FIVE REAL-ROBOT TASKS.

| Method | Charger | USB | Drawer | Wipe-ID | Wipe-OOD | Avg |
|---|---|---|---|---|---|---|
| DP | 20 | 15 | 60 | **95** | 0 | 38 |
| DP (force-concat) | 20 | 20 | 50 | 85 | 0 | 35 |
| RDP | 50 | 55 | 65 | 85 | 75 | 66 |
| **PhaForce** | **80** | **85** | **85** | **95** | **85** | **86** |

two external RealSense D435 cameras, and 80 teleoperated demonstrations per task collected using TactAR [18]. Diffusion runs at $f_s = 6$ Hz and control runs at $f_c = 24$ Hz with action chunk horizon $H_a = 16$. We compare against Diffusion Policy (DP) [1], DP with force concatenation, and RDP [18] on five real-robot tasks: *Charger Plug-in*, *USB Plug-in*, *Drawer Opening*, *Wiping (ID)*, and *Wiping (OOD)*. The OOD wiping setting raises the board by 3 cm at test time, creating an unseen contact geometry.

### B. Main results

Table I reports success rates over 20 real-robot trials per task, and Fig. 3 visualizes representative successful rollouts. **PhaForce** achieves the best or tied-best performance on all tasks, improving average success rate by 40 percentage points over the baselines. The gains are especially pronounced on plug-in tasks, where policies must switch among approach, planar search, recovery, and insertion. In Charger and USB plug-in, **PhaForce** improves SR from 50/55 to 80/85 over RDP, suggesting that explicit phase scheduling is critical when the policy must alternate between fine search, retreat-and-retry, and final insertion. In Drawer Opening, the gain is smaller but still consistent, indicating that phase-aware force usage also benefits constrained pulling under friction variation and occasional binding.

Compared with DP, **PhaForce** benefits from explicit force usage rather than purely visual inference of contact state. Compared with naive force concatenation, it uses force at the right timescale and in the right corrective subspaces. Compared with RDP, **PhaForce** provides an explicit phase schedule, which improves intent switching and reduces spurious high-rate corrections. This is particularly important in insertion, where timely search/recovery and subspace-restricted residuals improve fine alignment and reduce stagnation near the socket entrance. We further observe three characteristic failure modes in plug-in tasks for baseline methods: stagnation at the entrance, partial insertion without fully seating the connector, and slip-induced in-hand rotation caused by excessive rim contact. **PhaForce** mitigates the first two failure modes by coupling anticipatory contact/phase prediction with subspace-restricted residual correction, thereby enabling timely retreat-and-retry and more stable alignment.

For wiping, success rate alone is insufficient, because a policy may finish the motion while exhibiting unstable contact. On the in-distribution setting, **PhaForce** matches the strongest SR while improving contact quality. On the OOD setting, chunk-level planners without fast correction fail completely, whereas the slow–fast design remains feasible

TABLE II
WIPING QUALITY BEYOND SUCCESS RATE.

| Setting | Method | SR | Score | $\overline{F_n}$ | Over / Under |
|---------|--------|-----|-------|------|--------------|
| ID | RDP | 85 | 0.65 | 13.6 | 5.7% / 2.3% |
| | **PhaForce** | **95** | **0.85** | 12.3 | **4.5% / 2.0%** |
| OOD | RDP | 75 | 0.65 | 9.2 | 7.2% / **0.22%** |
| | **PhaForce** | **85** | **0.75** | 14.3 | **7.0%** / 0.34% |

TABLE III
ABLATION ON REPRESENTATIVE TASKS.

| Method | USB SR | Wipe-OOD SR | Wiping Score |
|--------|--------|-------------|--------------|
| w/o phase belief | 25 | 45 | 0.60 |
| w/o ORI | 35 | 60 | 0.45 |
| w/o Fast | 50 | 0 | – |
| **PhaForce** | **85** | **85** | **0.75** |

and **PhaForce** further improves over RDP, showing that phase-scheduled correction is critical for adapting to unseen contact geometry. This result is important because the OOD height shift is small in absolute magnitude but large enough to break purely demonstration-conditioned contact execution.

Table II further highlights that **PhaForce** improves wiping effectiveness and contact quality, not only binary success. In ID, it achieves a higher wiping score with lower over-/under-pressure ratio. In OOD, it improves both success rate and wiping score while maintaining comparable contact stability, indicating that explicit phase scheduling and high-rate correction remain effective under shifted contact geometry. This supports our central claim that force should not merely be fused into representation space, but should be scheduled in a task-phase-dependent manner across both planning and execution.

*C. Ablation*

Table III validates the major design choices. Removing phase belief severely degrades USB insertion, confirming that explicit scheduling is essential for routing corrections and triggering appropriate phase transitions. Replacing ORI with direct fusion reduces OOD wiping performance, supporting the role of ORI in preserving vision-dominant semantics while injecting force information. Removing the Fast corrector collapses OOD wiping, showing that high-rate residual correction is indispensable when the contact geometry deviates from demonstrations. Taken together, the ablations suggest that **PhaForce** does not derive its gains from any single added module. Instead, the benefit comes from coordinating *contact anticipation*, *phase-aware fusion*, and *control-rate correction* within a unified slow–fast design.

## IV. CONCLUSION

We presented **PhaForce**, a phase-scheduled visual–force policy that coordinates low-rate chunk planning and high-rate residual correction for contact-rich manipulation. The method uses CAP to predict an explicit contact/phase schedule, dual-gated visual–force fusion with ORI to preserve visual semantics in the Slow planner, and phase-routed residual correction in the Fast module. Experiments on five real-robot tasks show consistent gains in success rate, contact quality, and OOD robustness. These results suggest that explicit scheduling of *when / how much / where* to use force is a useful design principle for contact-rich visuomotor policy learning.

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
