# OpenReview forum: "PhaForce: Phase-Scheduled Visual–Force Policy Learning with Slow Planning and Fast Correction for Contact-Rich Manipulation"
_IEEE.org/ICRA/2026/Workshop/Manipulation_Robustness — ICRA 2026_

### Official Review · Reviewer_CG3u · 2026-05-05
**Review of Submission 32 by Reviewer CG3u**

**Rating:** 7
**Confidence:** 4

**Review:**

The paper proposes PhaForce, a slow–fast visuomotor policy for contact-rich manipulation that introduces an explicit contact/phase schedule to coordinate 1) a 6 Hz diffusion planner with Orthogonal Residual Injection (ORI) for vision–force fusion, and 2) a 24 Hz residual corrector that routes corrections to phase-specific subspaces (e.g., $(x,y,\text{yaw})$ for plug-in search; $z$ for wiping). A lightweight CAP head, trained anticipatorily on future-contact and future-phase labels, supplies the contact probability $p_t^c$ and phase belief $\mathbf{p}_t$ used by both stages. On five real-robot tasks (Charger, USB, Drawer, Wipe-ID, Wipe-OOD), PhaForce reaches 86% average SR vs. 66% (RDP) and 38% (DP), with notable OOD wiping robustness under a 3 cm board height shift.

Strengths

- Slow–fast separation is well-motivated. The 6 Hz / 24 Hz split with phase-routed residuals addresses a real timescale-mismatch gap in chunked diffusion policies, and the w/o Fast collapse on Wipe-OOD (0% SR) is a strong signal.
- ORI is a sensible mechanism. Projecting out the component parallel to $v_t$ before injection plausibly mitigates semantic drift; the OOD wiping ablation (w/o ORI: 60 → 85) supports the claim.
- Real-robot evaluation across diverse contact regimes (insertion, pulling, wiping) with an OOD condition and three baselines

Weaknesses

- Wiping Score is under-specified. Table II reports a "Score" column central to the wiping quality claims, but the paper does not define how it is computed.
- Phase labels are hand-defined per task. $K$ task-specific phases are assumed but the paper does not discuss how phases were chosen, label cost, or sensitivity to $K$. This limits the "design principle" framing in the conclusion.
- Ablations are partial. No ablation isolates the head-wise gate $g_t^{\text{head}}$ vs. the global gate $g_t^c$, nor the diagonal mask routing vs. a learned dense gating. The "w/o phase belief" row collapses several mechanisms at once.
- Baseline scope. No comparison against admittance/compliance-control hybrids (e.g., ACP [14]) on the same tasks — these are the closest competitors for the "where to use force" question.

Minor

- Eq. (4) uses $\alpha$ as an injection scale but $\alpha$ is not specified.
- Eq. (2) better explains the symbols even though reader can infer they are ground truth and predicted logits.

---

### Decision · Program_Chairs · 2026-05-21

Accept